# Potential Efficacy of Propolis in Treating *Helicobacter pylori* Infection and Its Mechanisms of Action

**DOI:** 10.3390/nu17172803

**Published:** 2025-08-28

**Authors:** Haitao Nie, Qing Li, Keke Zhao, Wen Li, Cuiping Zhang, Xiasen Jiang

**Affiliations:** 1Biology and Food Engineering School, Fuyang Normal University, Fuyang 236000, China; nieht@fynu.edu.cn (H.N.); liqing@stu.fynu.edu.cn (Q.L.); 17856785061@163.com (K.Z.); 13008557175@163.com (W.L.); 2College of Animal Sciences, Zhejiang University, Hangzhou 310058, China

**Keywords:** *Helicobacter pylori*, propolis, antibacterial, immunomodulation, anti-inflammatory, antioxidant, gastrointestinal diseases

## Abstract

Background: *Helicobacter pylori* (*H. pylori*) is a major pathogen associated with a variety of gastrointestinal disorders, including gastritis, peptic ulcers, and gastric cancer. As a natural bioactive product, propolis exhibits multifaceted and multi-mechanistic effects. Due to its immunomodulatory, anti-inflammatory, and antioxidant properties, propolis has emerged as a promising therapeutic alternative, offering an innovative approach to managing *H. pylori* infections and providing new insights into addressing antibiotic resistance. Methods: This comprehensive review, synthesizing data from PubMed, ScienceDirect, and SciFinder, examines the mechanisms by which propolis combats *H. pylori*. Results: Propolis has demonstrated significant antibacterial efficacy against *H. pylori* in both in vitro and in vivo models. Its multitargeted mechanisms of action include direct inhibition of bacterial growth, interference with the expression of virulence factors, suppression of virulence-associated enzymes and toxin activity, immunomodulation, and anti-inflammatory effects. These combined actions alleviate gastric mucosal inflammation and damage, reduce bacterial colonization, and promote mucosal healing through antioxidant and repair-promoting effects. Furthermore, propolis disrupts oral biofilms, restores the balance of the oral microbiome, and exerts bactericidal effects in the oral cavity. Synergistic interactions between propolis and conventional medications or other natural agents highlight its potential as an adjunctive therapy. Conclusions: Propolis demonstrates dual functionality by inhibiting the release of inflammatory mediators and suppressing *H. pylori* growth, highlighting its potential as an adjuvant therapeutic agent. However, clinical translation requires standardized quality control and higher-level clinical evidence. Future research should focus on validating its clinical efficacy and determining optimal dosing regimens, and exploring its role in reducing *H. pylori* recurrence.

## 1. Introduction

*Helicobacter pylori* (*H. pylori*) is a Gram-negative, spiral-shaped microaerobic bacterium that uniquely colonizes the highly acidic human stomach, representing the sole known microorganism adapted to this inhospitable niche. Chronic *H. pylori* infection is a well-established etiological agent for a spectrum of upper gastrointestinal diseases, including chronic active gastritis, peptic ulcer disease, gastric mucosa-associated lymphoid tissue (MALT) lymphoma, and gastric adenocarcinoma, particularly of the non-cardia subtype. Consequently, it has been classified as a Group 1 carcinogen by the World Health Organization [1,2,3,4].

The current first-line therapy for *H. pylori* infection is quadruple therapy, typically comprising a proton pump inhibitor (PPI) and a bismuth salt, combined with two antibiotics such as amoxicillin, clarithromycin, metronidazole, tetracycline, or levofloxacin [5]. However, the efficacy of these antibiotic-dependent regimens is increasingly compromised by rising antimicrobial resistance. Global resistance rates to key antibiotics, notably clarithromycin and levofloxacin, have led to eradication rates that widely fluctuate (65–92%) and are declining in many regions. In response to this escalating public health threat, the WHO has designated *H. pylori* as a high-priority antibiotic-resistant pathogen, underscoring the urgent need for non-antibiotic alternatives or adjunctive therapies [6]. Furthermore, current treatments are frequently associated with adverse effects, including nausea, vomiting, headache, dysgeusia, dizziness, and gut microbiota dysbiosis. These side effects adversely impact patient tolerance and adherence, ultimately compromising therapeutic outcomes [5,7].

Against this backdrop, the discovery of highly effective, low-toxicity antimicrobial agents from natural sources has emerged as a critical research direction for combating *H. pylori* infections. Propolis, a natural resinous substance produced by honeybees, has a long history of use in traditional medicine, with records dating back to ancient Egyptian and Greek civilizations [8]. Its chemical composition is remarkably complex, comprising over 20 major classes of compounds and more than 500 identified constituents, including a diverse array of bioactive molecules such as flavonoids, phenolic acids, terpenoids, and esters [9]. This composition is highly variable and depends significantly on geographical origin, botanical source, and seasonal factors. It is this profound chemical diversity that underpins the multifaceted biological activities of propolis. Modern pharmacological studies have confirmed that propolis exhibits broad-spectrum antimicrobial, potent anti-inflammatory, robust antioxidant, immunomodulatory, and mucosal protective properties [10]. Consequently, owing to its pronounced efficacy against *H. pylori* and biofilms, coupled with its capacity to mitigate gastric mucosal inflammation, propolis has garnered considerable interest as a promising alternative or adjunctive therapy for *H. pylori*-associated infections and gastric pathologies [11]. Investigating the therapeutic potential of propolis against *H. pylori* not only offers a strategy to circumvent the challenge of antibiotic resistance but also holds promise for reducing inflammatory damage to the gastric mucosa and enhancing patient tolerance through its multi-targeted mechanisms of action. Such research provides a compelling scientific foundation for the development of next-generation therapeutic formulations.

## 2. The Pathogenic Mechanisms of *H. pylori* Infection

The pathogenesis of *H. pylori* is a complex, multi-factorial process involving a dynamic interplay of mechanisms that alter the gastric microenvironment, induce inflammation, exert direct cytotoxicity, facilitate immune evasion, and cause genotoxicity. During initial colonization, the bacterium utilizes urease to neutralize gastric acid and disrupts the mucosal barrier, establishing a niche for persistent infection. This chronic infection state often leads to hypochlorhydria, which further perturbs the gastric microbiota. Key virulence factors, such as CagA and VacA, not only inflict direct epithelial injury and vacuolation but also potently amplify inflammatory cascades by activating signaling pathways like NF-κB and AP-1, thereby promoting cytokine release and immune cell infiltration. Furthermore, *H. pylori* employs sophisticated strategies for immune evasion, including antigenic variation, immunosuppression, and metabolic interference, enabling it to evade host clearance and establish persistent infection. The resultant chronic inflammation and oxidative stress induce DNA damage and impair repair mechanisms, ultimately culminating in a significantly elevated risk of gastric carcinogenesis. These interconnected mechanisms collectively drive the progression of *H. pylori*-associated gastric pathology along the Correa cascade—from inflammation and atrophy to intestinal metaplasia and dysplasia, and finally to malignancy. A comprehensive understanding of these intricate pathogenic pathways is therefore paramount for elucidating the disease process and for developing novel preventive and therapeutic interventions.

### 2.1. Destruction of the Gastric Mucosal Barrier

*H. pylori* infection disrupts key gastric environmental factors, including gastric acid secretion and the mucosal barrier, in a stage-dependent manner that is influenced by the host immune response [12]. In the initial phase of infection, the bacterial virulence factor urease catalyzes the hydrolysis of urea into ammonia and carbon dioxide. The ammonia neutralizes gastric acid, creating a neutral microenvironment conducive to bacterial colonization, while the carbon dioxide contributes to a transient stimulation of acid secretion [13,14]. The ammonia not only shields *H. pylori* from gastric acid but also inflicts direct damage on gastric epithelial cells, thereby compromising the integrity and defensive functions of the mucosal barrier [15]. This breach in mucosal integrity allows gastric acid and digestive enzymes to penetrate the submucosa, triggering a cascade of inflammatory reactions and tissue injury. As the infection progresses to a chronic state, persistent inflammation and mucosal damage impair the function of gastric parietal cells, leading to hypochlorhydria (reduced acid secretion) and a consequent rise in gastric pH. This low-acid environment not only favors the persistence of *H. pylori* but also facilitates the aberrant colonization of non-*H. pylori* bacteria, increasing the risk of gastritis, peptic ulcers, and gastric cancer [16,17]. Beyond direct damage, *H. pylori* disrupts the balance of key gastrointestinal hormones, such as gastrin and somatostatin, and interferes with the mucosa’s intrinsic feedback regulatory mechanisms, further perturbing gastric acid production [18]. Furthermore, the continuous accumulation of ammonia exacerbates mucosal structure damage, ultimately creating a microenvironment that sustains chronic inflammation and aberrant epithelial hyperplasia, playing a central role in the pathogenesis of *H. pylori*-associated gastric diseases (Figure 1A) [19,20].

### 2.2. Facilitation of Inflammatory Responses

*H. pylori* activates host macrophages and gastric epithelial cells through its cell wall components (such as lipopolysaccharides and peptidoglycans) and various toxic molecules, inducing the release of inflammatory mediators such as tumor necrosis factor-α (*TNF-α*), interleukin (*IL-1β*, *IL-6*, *IL-8*), thereby initiating the innate immune response [21]. In these processes, the nuclear factor-κB (NF-κB) signaling pathway plays a central regulatory role. Following its injection into host cells via the type IV secretion system (T4SS), the CagA protein activates the IκB kinase (IKK) complex. This leads to the phosphorylation and ubiquitin-mediated degradation of its inhibitor, IκBα, which in turn promotes the nuclear translocation of NF-κB dimers (e.g., p50–p65). Upon entering the nucleus, NF-κB binds to specific κB sites, initiating the transcription of various pro-inflammatory genes, including TNF-α, IL-1β, IL-6, IL-8, COX-2, and iNOS [22,23,24]. Additionally, *H. pylori* infection can activate the MAPK signaling pathway, further promoting the activation of the transcription factor AP-1. AP-1 and NF-κB not only exhibit synergistic cross-talk at the transcriptional level but also jointly upregulate the expression of matrix metalloproteinases (MMPs), adhesion molecules (such as ICAM-1), and chemokines. This synergistic action significantly enhances the infiltration of neutrophils and monocytes, amplifying the local inflammatory and tissue destructive within the gastric mucosa (Figure 1B) [25,26]. Additionally, sustained activation of inflammatory signals can inhibit apoptosis of gastric epithelial cells and promote aberrant proliferation, contributing to the formation of a microenvironment conducive to precancerous lesions.

### 2.3. Pathogenic Effects of Cytotoxins

*H. pylori* can survive stably in the harsh environment of the stomach, primarily relying on a variety of virulence factor repositories, including *CagA*, *VacA*, and the type IV secretion system (T4SS). These factors act synergistically to cause a range of gastric pathological changes, spanning from chronic gastritis and peptic ulcers to mucosa-associated lymphoid tissue lymphoma and gastric cancer [27,28]. Among these, the *VacA* toxin forms anion channels on the host cell membrane, disrupting mitochondrial membrane potential and inducing cytoplasmic vacuolization, ultimately leading to cell death [29]. The *CagA* protein is transported into host cells via the type IV secretion system, interfering with multiple intracellular signaling pathways and thereby enhancing cellular proliferation and inflammatory responses [30]. The synergistic effects of these factors collectively encode the core pathogenic strategy of *H. pylori*: the direct toxic effects of *CagA* and the injection of additional substances further exacerbate cellular damage and inflammatory states, continuously driving the progression of infection-related gastric mucosal pathology [31]. Additionally, bacterial outer membrane proteins (such as OipA, HopQ, and HopZ) act as important pro-inflammatory molecules, effectively stimulating inflammatory responses in gastric epithelial cells; while proteins encoded by the *IceA* and *dupA* genes are closely associated with the pathogenesis of specific clinical outcomes (such as duodenal ulcers or gastric cancer) (Figure 1C) [32,33,34].

### 2.4. Immune Evasion

To achieve long-term colonization in the host’s gastric mucosa, *H. pylori* has evolved multi-layered immune evasion strategies encompassing multiple stages of both innate and adaptive immunity. Its surface antigens, such as the O antigen chain structure of lipopolysaccharide (LPS), exhibit high variability, enabling them to mimic the glycosaminoglycans on the host cell surface, thereby reducing immunogenicity and evading recognition by Toll-like receptors (TLRs) [35,36]. Additionally, outer membrane proteins (such as BabA and SabA) regulate their expression through antigenic drift and phase change, thereby evading neutralization by host antibodies [37]. In terms of immune suppression, virulence factors secreted by *H. pylori*, such as *CagA* and *VacA*, can interfere with T cell responses. *CagA* can inhibit the T cell receptor signaling pathway and block the differentiation of Th1 and Th17 cells, while *VacA* can induce the formation of mitochondrial membrane channels, inhibit T cell proliferation, and promote the expansion of regulatory T cells (Tregs), thereby weakening effective immune clearance [38]. Bacteria can also deplete glutamine in the microenvironment by synthesizing γ-glutamyl transpeptidase (GGT), thereby inhibiting T cell metabolic activation and inducing macrophages to polarize toward a tolerant phenotype [39]. These mechanisms collectively contribute to the establishment of a chronic infection state in the gastric mucosa that is difficult to clear. Persistent low-grade immune activation and a locally suppressive microenvironment not only facilitate bacterial persistence but also promote inflammatory responses, thereby creating conditions conducive to the development of gastric mucosal lesions such as atrophy, intestinal metaplasia, and even gastric cancer (Figure 1D) [40].

### 2.5. Genotoxic Effects

During *H. pylori* infection, the production of genotoxic substances such as reactive oxygen species (ROS) and reactive nitrogen species (RNS) significantly increases due to extensive infiltration of inflammatory cells and epithelial cell stress. These molecules can cause various types of DNA damage, including base modifications, single-strand or double-strand breaks, and chromosomal aberrations, severely disrupting genomic stability [41]. Persistent infection and inflammation can also promote malignant transformation of gastric epithelial cells through multiple mechanisms, including epigenetic alterations, activation of proto-oncogenes, and inactivation of tumor suppressor genes [42]. Numerous clinical and experimental studies have shown that levels of DNA damage biomarkers in gastric mucosal tissue are significantly elevated in individuals infected with *H. pylori*, and these levels are positively correlated with cancer risk [43]. During the abnormal proliferation and repair processes driven by chronic inflammation, DNA damage continues to accumulate while repair efficiency decreases, further increasing the probability of carcinogenic mutations. Therefore, the persistent inflammatory and oxidative stress microenvironment created by *H. pylori* infection is considered a key driving factor in gastric cancer development (Figure 1E) [44].

## 3. The Chemical Composition of Propolis

Propolis is a natural resinous substance formed by honeybees from the exudates of plant buds and tree bark, combined with secretions from their salivary glands. It exhibits a broad spectrum of pharmacological properties, including potent antibacterial, anti-inflammatory, and antioxidant activities, as well as the capacity to modulate immune responses and the gut microbiota [45]. These diverse bioactivities are attributed to its highly complex and heterogeneous chemical composition. Rather than relying on a single active constituent, propolis exerts its effects through the synergistic interplay of more than 1000 identified compounds [46]. This complex mixture comprises a wide array of chemical classes, such as flavonoids, terpenoids, phenolic acids and their esters, quinones, vitamins, and trace elements, which collectively constitute the material foundation for its multifaceted pharmacological functions [47,48].

### 3.1. Chemical Composition and Structure of Propolis

The flavonoid profile of propolis is remarkably diverse, comprising 11 distinct subclasses: flavones, flavonols, flavanones, flavanols, isoflavones, dihydroisoflavones, flavans, isoflavans, chalcones, dihydrochalcones, and neoflavonoids (Figure 2) [35]. To date, nearly 300 flavonoids have been identified in propolis, which predominantly exist as aglycones and their methoxy derivatives. Several of these compounds, including apigenin, chrysin, luteolin, kaempferol, quercetin, galangin, pinocembrin, naringenin, and acacetin-3-ethyl ester, have demonstrated significant biological activity [49].

Propolis is also rich in phenolic acids, with over a hundred compounds identified to date. These are primarily classified into two structural scaffolds: the C6-C1 type, comprising benzoic acid derivatives (e.g., gallic, protocatechuic, vanillic, and gentisic acids), and the C6-C3 type, which includes the phenylpropanoids [45]. A prominent subgroup within the C6-C3 category is the hydroxycinnamic acids, featuring compounds such as cinnamic, *p*-coumaric, caffeic, and ferulic acids, as well as the notable artepillin C. This subgroup also encompasses various derivatives and esters, including 3,4-dimethoxycinnamic acid, *p*-coumaric acid benzyl ester, and the bioactive compound caffeic acid phenethyl ester (CAPE) (Figure 3) [50]. Furthermore, certain types of propolis, notably Brazilian green propolis, are characterized by the presence of chlorogenic acid compounds, including chlorogenic acid, 4,5-dicaffeoylquinic acid, 3,5-dicaffeoylquinic acid, and 3,4,5-tricaffeoylquinic acid [51].

Propolis is a rich source of terpenes, with over 200 identified constituents. Based on their structural skeletons, these terpenes are classified into four major categories: monoterpenes (C10), sesquiterpenes (C15), diterpenes (C20), and triterpenes (C30), which are distinguished by the number of isoprene units. Monoterpenes exhibit a variety of structural frameworks, including acyclic, monocyclic, and bicyclic forms, frequently featuring oxygen-containing functional groups (Figure 4A). Sesquiterpenes are characterized by their structural diversity, encompassing acyclic, monocyclic, bicyclic, and tricyclic skeletons (Figure 4B). Diterpenes are represented by mono-, bi-, tri-, and tetracyclic structures (Figure 4C). In contrast, triterpenes are predominantly found as tetracyclic and pentacyclic derivatives (Figure 4D) [52,53,54].

### 3.2. Biological Activity of Propolis

The diverse biological activities of propolis are directly attributed to its rich natural bioactive components. These components exhibit a wide range of pharmacological potential through complex and synergistic mechanisms of action, including significant antibacterial, anti-inflammatory, and antioxidant effects. Its antibacterial activity is particularly notable, capable of inhibiting a variety of Gram-positive bacteria, Gram-negative bacteria, certain fungi, and even some viruses [55]. The primary mechanisms involve flavonoids (such as poplarin and pinobanksin), phenolic acids, and terpenoid compounds interfering with microbial cell membrane structure, energy metabolism, and biofilm formation [56]. In terms of anti-inflammatory effects, propolis regulates key signaling pathways such as NF-κB and MAPK to inhibit the release of pro-inflammatory factors like TNF-α and IL-6, thereby alleviating acute and chronic inflammatory responses [57]. Additionally, its high content of polyphenols and flavonoids confers exceptional antioxidant capacity, effectively scavenging free radicals and reducing oxidative stress damage, offering potential value in delaying cellular aging and preventing oxidative-related diseases [58]. In the digestive system, Brazilian green propolis demonstrates significant efficacy in regulating intestinal microbiota dysbiosis and treating inflammatory bowel disease, showing great potential as an adjunctive therapy for gastrointestinal disorders [59].

## 4. Mechanisms of Propolis Action Against *H. pylori*

Propolis combats *H. pylori* infection through its diverse biological activities, acting in a synergistic manner via multiple targets and pathways. It not only directly affects bacteria by inhibiting their growth, adhesion, and expression of virulence factors, but also regulates the host immune response, alleviating inflammation and oxidative stress, thereby enhancing gastric mucosal barrier function and promoting ulcer healing (Figure 5). The following sections provide a detailed explanation of its mechanisms of action from multiple perspectives (Table 1).

### 4.1. Antibacterial Activity

Propolis exhibits significant antibacterial activity against *H. pylori*, a property substantiated by extensive research across diverse geographical regions, including China [60,61], Brazil [62,63], Georgia [64], Italy [65], Turkey [66,67,68], Bulgaria [69,70], Chile [71], Portugal [72], South Korea [73], Kazakhstan [74], Indonesia [75], Poland, Ukraine, Greece [76] and Iran [77]. Studies employing various evaluation methods, including the agar hole diffusion method, agar dilution method, and disk diffusion method, have demonstrated that propolis extracts exhibit strong dose-dependent antibacterial activity against *H. pylori*. For example, research by Baltas et al. revealed that ethanolic extracts of 15 different propolis samples produced inhibition zones ranging from 31.0 to 47.0 mm in diameter against the *H. pylori* J99 strain [66]. Further confirming this dose–response relationship, Ibrahim and Turab demonstrated a strong positive correlation between propolis concentration and its inhibitory effect using mucosal biopsy specimens from gastric ulcer patients, yielding findings of significant clinical relevance [65]. Critically, the antibacterial efficacy of propolis is maintained under simulated gastric conditions, proving effective against *H. pylori* strains harboring multiple virulence factors [78]. Notably, propolis displays synergistic potential when combined with conventional antibiotics. For example, the combination of propolis extract with clarithromycin significantly enhances antibacterial activity against clarithromycin-sensitive, cagA-positive *H. pylori* strains [79]. Furthermore, synergistic effects are also observed when propolis is paired with extracts from traditional Chinese medicinal herbs, including Coptis, Evodia, Prunus, Lonicera, and Ginkgo. Specifically, combinations with Evodia or Prunus result in a marked enhancement of antibacterial potency [80,81].

The primary mechanism underlying the antibacterial action of propolis is the disruption of bacterial cell membrane integrity and permeability [82]. Key bioactive constituents—namely phenolic compounds, terpenoids, and aromatic acids—compromise the structural integrity of the cell membrane. This disruption leads to the leakage of intracellular contents, ultimately causing cell lysis and bacterial death [83]. This membrane-damaging mechanism not only is directly bactericidal but may also facilitate the increased intracellular penetration of co-administered antibiotics, thereby potentially overcoming bacterial resistance and reducing the likelihood of its development [62,84].

### 4.2. Inhibition of Virulence-Associated Enzymes

Propolis exerts significant inhibitory effects on several key enzymes critical for the pathogenicity and survival of *H. pylori*. A primary target is bacterial urease, an enzyme essential for neutralizing gastric acid and establishing colonization [85]. Ethanol extracts of propolis have demonstrated potent inhibition of *H. pylori* urease, with half-maximal inhibitory concentration (IC_50_) values ranging from 0.260 to 1.525 mg/m [66]. Mechanistic studies employing molecular docking simulations and in vitro colorimetric assays have identified the 3-OH, 5-OH, and 3′,4′-dihydroxy moieties on flavonoids as crucial pharmacophores responsible for this inhibitory activity [78,86]. Furthermore, propolis targets peptidyl deformylase (*H. pylori*-PDF), a metalloenzyme vital for bacterial protein maturation. *H. pylori*-PDF catalyzes the removal of the N-formyl group from nascent polypeptides, a process indispensable for bacterial viability and virulence [87]. Cui et al. demonstrated that CAPE acts as a competitive inhibitor of *H. pylori*-PDF, significantly impairing the survival capacity of *H. pylori* [88]. Notably, CAPE binds to the enzyme’s active site without interacting with the catalytic Co^2+^ ion, and this unique binding mechanism may reduce potential side effects on human metalloproteases [89].

Beyond these direct antibacterial targets, propolis also modulates host inflammatory pathways. *H. pylori* infection induces the upregulation of cyclooxygenase-2 (COX-2), an enzyme that exacerbates gastric mucosal inflammation. CAPE has been shown to suppress this COX-2 induction, thereby exerting significant anti-inflammatory and chemopreventive effects [90,91,92,93]. Additionally, propolis inhibits xanthine oxidase (XO), an enzyme responsible for generating reactive oxygen species (ROS). By curbing XO activity, propolis reduces ROS production, thereby protecting the gastric mucosa from oxidative stress and associated damage [94,95,96].

### 4.3. Immune Regulation and Anti-Inflammatory Effects

Propolis possesses significant immunomodulatory and anti-inflammatory properties, providing a crucial mechanistic basis for its therapeutic potential in *H. pylori* infections. In a gastric adenocarcinoma cell model of *H. pylori* infection, propolis reduces the expression levels of various pro-inflammatory cytokines (including IL-8, IL-12, IL-1β) and inflammatory markers (such as TNF-α, COX-2, iNOS) [97,98,99]. This anti-inflammatory effect is primarily achieved by inhibiting the MAPK and NF-κB signaling pathways [100,101]. Specific components in propolis, such as CAPE, exhibit distinct immunomodulatory effects. It interferes with T-cell receptor-mediated activation processes, inhibits T-cell activity, and regulates the ERK signaling pathway [102]. Studies have shown that CAPE can inhibit the activation of NF-κB and AP-1, as well as the secretion of IL-8 and TNF-α, in gastric adenocarcinoma cells induced by *H. pylori* [103]. By preventing the degradation of IkB-α protein, CAPE can inhibit the nuclear translocation of NF-κB/p65, thereby suppressing *H. pylori*-induced cell proliferation and inflammatory responses [104].

In addition, propolis has significant antioxidant properties. It effectively scavenges free radicals, reduces oxidative stress damage, and enhances cellular antioxidant defense capabilities by activating the Nrf2 pathway [105,106]. This effect leads to the upregulation of antioxidant enzymes such as superoxide dismutase (SOD) and glutathione peroxidase (GPx), further strengthening the cell’s resistance to oxidative damage [101].

### 4.4. In Vivo Suppression of H. pylori

Multiple animal studies have confirmed the inhibitory effect of propolis on *H. pylori* in vivo. Korean propolis demonstrated significant anti-inflammatory and antibacterial effects in a mouse model of gastric mucosal damage induced by *H. pylori* [100,107]. It can effectively inhibit the proliferation of *H. pylori* and reduce the expression of various virulence factors (including *CagA*, urease A, surface antigens, and neutrophil-activating protein A) [108]. The ethanol extract of Brazilian red propolis (300 mg/kg) significantly reduced the *H. pylori* load in rat gastric mucosa and regulated inflammatory responses [63]. Interestingly, studies have found that combining *OipA* (an essential adhesin of *H. pylori*) with propolis, especially after oral administration (gastrointestinal administration), can increase interferon-γ (IFN-γ) expression by 11-fold, suggesting that propolis may serve as an effective oral adjuvant to enhance the immunogenicity of vaccine antigens [109,110].

The combined use of propolis with other natural extracts demonstrates superior efficacy. Studies indicate that propolis complexes, when used in combination with olive leaf and licorice extracts, more effectively inhibit *H. pylori*-induced gastric mucosal ulcers, reduce bacterial colonization, and alleviate gastric mucosal inflammation [111,112,113]. Compared to the use of propolis alone, the complex exhibits stronger efficacy in inhibiting *H. pylori* adhesion, antioxidant activity, and anti-inflammatory effects [114,115]. Additionally, the combination of propolis with probiotics also demonstrates good synergistic effects. In a study using 54 male Wistar rats (weighing 200–250 g), the animals were inoculated with an *H. pylori* suspension (10^8^ CFU/mL) and then administered propolis, probiotics (both at a concentration of 10^8^ CFU/mL), or a combination of both via gavage for 21 days. The experimental results showed that the propolis and probiotics combination therapy group exhibited the best overall effect, indicating that these two formulations have complementary effects in regulating the gastrointestinal microbiota and combating *H. pylori* [116,117]

Clinical study results indicate that propolis demonstrates certain potential against *H. pylori* in humans, but its efficacy is significantly influenced by formulation type and treatment regimen. A trial involving 18 infected participants showed that after seven consecutive days of oral administration of Brazilian green propolis extract (3 times daily, 20 drops per dose), 50% of participants experienced a decrease of over 20% in their urea breath test values, indicating an initial inhibitory effect; However, after a 40-day treatment cycle, 83% of participants failed to achieve complete eradication of *H. pylori*, suggesting that while propolis monotherapy has some activity, it is insufficient for effective eradication and may require optimization of the dosing regimen, such as increasing the dose or frequency, to achieve sufficient antimicrobial concentrations [118]. On the other hand, a comparative study revealed more promising results: propolis tablets showed a trend toward superiority over conventional triple therapy (colloidal bismuth subsalicylate combined with amoxicillin and famotidine) in terms of *H. pylori* eradication, suggesting that optimizing propolis formulations or combining it with other medications may be effective strategies to enhance its clinical efficacy [119,120,121].

### 4.5. Effects on Oral H. pylori

The oral cavity serves as an important reservoir for *H. pylori* and plays a crucial role in the transmission and recurrence of *H. pylori* infection [122]. The efficacy of propolis in eliminating *H. pylori* in the oral cavity has been confirmed by multiple studies. Clinical research indicates that using mouthwash containing propolis can significantly reduce the load of *H. pylori* in dental plaque and saliva, thereby lowering the risk of cross-infection between the oral cavity and the stomach [123].

The mechanism by which propolis inhibits *H. pylori* in the oral cavity involves multiple aspects. First, the flavonoids and phenolic acids in propolis can disrupt the integrity of bacterial cell membranes, thereby inhibiting the adhesion ability and biofilm formation of *H. pylori* [124]. Second, propolis can competitively inhibit the colonization of *H. pylori* by regulating the balance of the oral microbiota. Studies have found that propolis can selectively inhibit the growth of pathogenic bacteria while having little effect on beneficial bacteria that are normally present in the oral cavity. This selective antibacterial property is beneficial for maintaining the balance of the oral microecology [125].

It is worth noting that the efficacy of propolis in eliminating *H. pylori* in the oral cavity is closely related to its formulation. Propolis oral spray can directly target areas where *H. pylori* hides, such as periodontal pockets and mucosal folds, thereby increasing local drug concentration; whereas propolis lozenges can prolong the duration of drug action and enhance antibacterial efficacy [79,126]. Clinical comparative studies have shown that the use of propolis-containing oral care products in combination with systemic therapy can increase the eradication rate of *H. pylori* by 15–20% and significantly reduce the reinfection rate [126]. Additionally, propolis can improve oral clinical manifestations associated with *H. pylori* infection. After using propolis formulations, patients not only experience a decrease in *H. pylori* load but also significant improvements in oral health indicators such as periodontal inflammation indices and gingival bleeding indices. This further supports the application value of propolis in comprehensive *H. pylori* treatment, particularly in interventions targeting the oral-gastric transmission route [125].

**Table 1 nutrients-17-02803-t001:** Anti-*H. pylori* Activities of Propolis: Mechanisms and Evidence Across Study Models.

Propolis Type/Source	Research Model	Major Method	Key Findings	Ref.
Multiple types of propolis	In vitro/molecular simulation	Agar diffusion method, dilution method	Dose-dependent antibacterial activity against *H. pylori*; Suppression zone of 31.0–47.0 mm	[60,61,62,63,64,65,66,67,68,69,70,71,72,73,74,75,76,77]
Multi country propolis extract	In vitro	Cell membrane permeability measurement; molecular docking	Disrupting the structure of bacterial cell membranes and promoting the entry of antibiotics; Flavonoids are key hydroxyl groups that inhibit urease (IC_50_: 0.260–1.525 mg/mL)	[66,78,86]
Prololis With clarithromycin or TCM herbs	In vitro (combination)	combined chemosensitivity test	Synergistic effects with antibiotics (e.g., clarithromycin) and herbal extracts (e.g., Evodia, Prunus)	[79,80,81]
Propolis + Probiotics	In vivo (rat)	21-day combination therapy	Synergistic promotion of weight gain, improvement of gastric tissue pathology, reduction in inflammation and cell apoptosis	[116,117]
Propolis + OipA protein	In vivo	Oral immunization model	11-fold increase in IFN-γ expression; potential oral adjuvant effect	[109,110]
Korean propolis	In vivo (mouse)	*H. Pylori* infection model	Reduce bacterial load and expression of virulence factors (CagA, UreA) in the stomach, alleviate inflammation and mucosal damage	[100,107,108]
Brazilian red propolis Ethanol extract	Rat model	Ethanol extract treatment (300 mg/kg)	Significantly reduce the area of gastric ulcer (49.4%), enhance antioxidant enzyme activity, and promote mucosal repair	[63]
Brazilian green propolis	Clinical	Oral drops, 7-day treatment	50% of subjects showed >20% UBT decrease; 83% not eradicated after 40 days	[118]
Propolis tablets	Clinical	Comparison with Triple Therapy	The clearance rate shows a trend superior to conventional triple therapy	[119,120,121]
Propolis mouthwash/spray	Clinical	Mouthwash/spray combined system treatment	Reduce oral *H. pylori* load, increase overall eradication rate by 15–20%, and improve periodontal indicators	[123,126]
Propolis (CAPE)	In vitro enzymology	Enzyme activity inhibition	Competitive inhibition of *H. pylori* peptide deacetylase (PDF), unique binding mechanism reduces side effects on human enzymes	[88,89]
	Cell model	ELISA, Western blot	Inhibit the NF–κB and MAPK pathways, reduce the expression of TNF–α, IL-8, COX-2, etc.	[97,98,99,100,101,103]

## 5. Conclusions

Propolis, as a natural bioactive product, exhibits multifaceted and multi-mechanistic effects in combating *H. pylori* infection. It not only directly inhibits bacterial growth, interferes with the expression of virulence factors, and suppresses toxin activity but also modulates host immune inflammatory responses, enhances antioxidant defense mechanisms, and promotes mucosal repair capabilities, thereby exerting a comprehensive therapeutic effect. Its diverse chemical composition and multi-targeted bioactive properties offer new insights into addressing antibiotic resistance in *H. pylori*, particularly when combined with probiotics, herbal extracts, or other antibiotics, where synergistic effects are demonstrated. However, the clinical application of propolis still faces numerous challenges, including its complex chemical composition, significant variability in source, and substantial fluctuations in active component content. Standardized production processes and quality control systems are urgently needed to ensure the stability and reproducibility of therapeutic effects. Additionally, most current research remains at the in vitro and animal experiment stage, with limited clinical evidence, especially large-scale, multi-center randomized controlled trials. Future research should focus more on the separation and identification of propolis’ active components, the in-depth analysis of its mechanisms of action, the optimization of clinical dosing regimens (including formulation design, dose exploration, and determination of treatment duration), and its practical value as an adjunctive therapeutic strategy in the eradication of *H. pylori* and the reduction of recurrence. Through systematic and in-depth exploration, propolis holds promise as a natural agent with broad application potential for *H. pylori* eradication, offering new strategies and options for the prevention and treatment of *H. pylori* infection and related gastric diseases.

## Figures and Tables

**Figure 1 nutrients-17-02803-f001:**
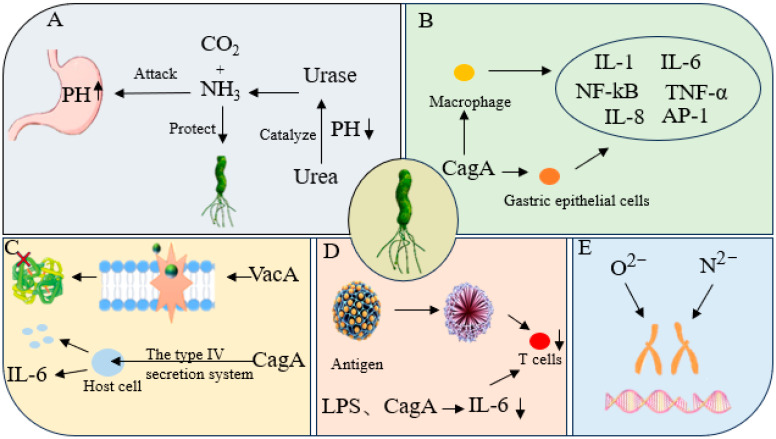
The pathogenic mechanisms of *H. pylori* infection. (**A**) The mechanisms of gastric mucosal barrier disruption and promotion of inflammatory response; (**B**) The mechanism of regulating gastric acid secretion; (**C**) The mechanism of pathogenic effects of cytotoxins; (**D**) The mechanism of immune evasion; (**E**) The mechanism of genotoxic effects. (Arrows indicate an increase or decrease in pH or activity, while X represents the destruction of proteins).

**Figure 2 nutrients-17-02803-f002:**
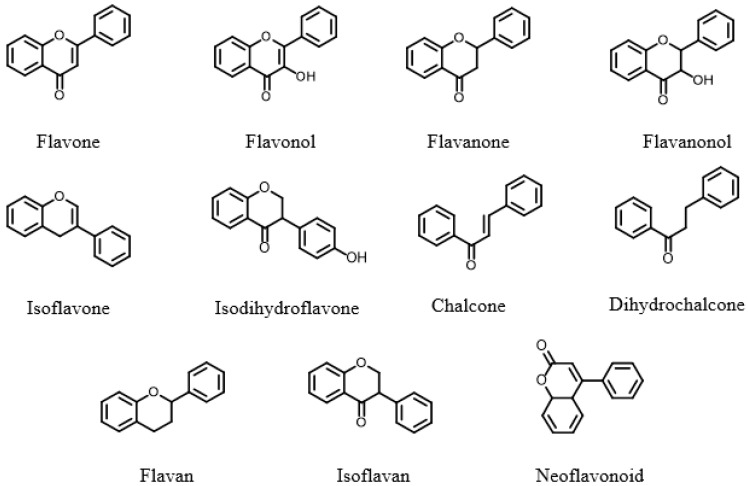
Types of Flavonoid Compounds in Propolis.

**Figure 3 nutrients-17-02803-f003:**
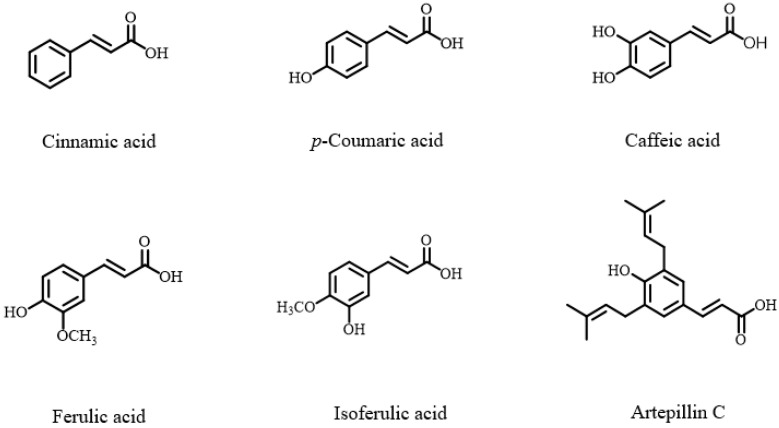
Chemical structure of major phenolic acids in propolis.

**Figure 4 nutrients-17-02803-f004:**
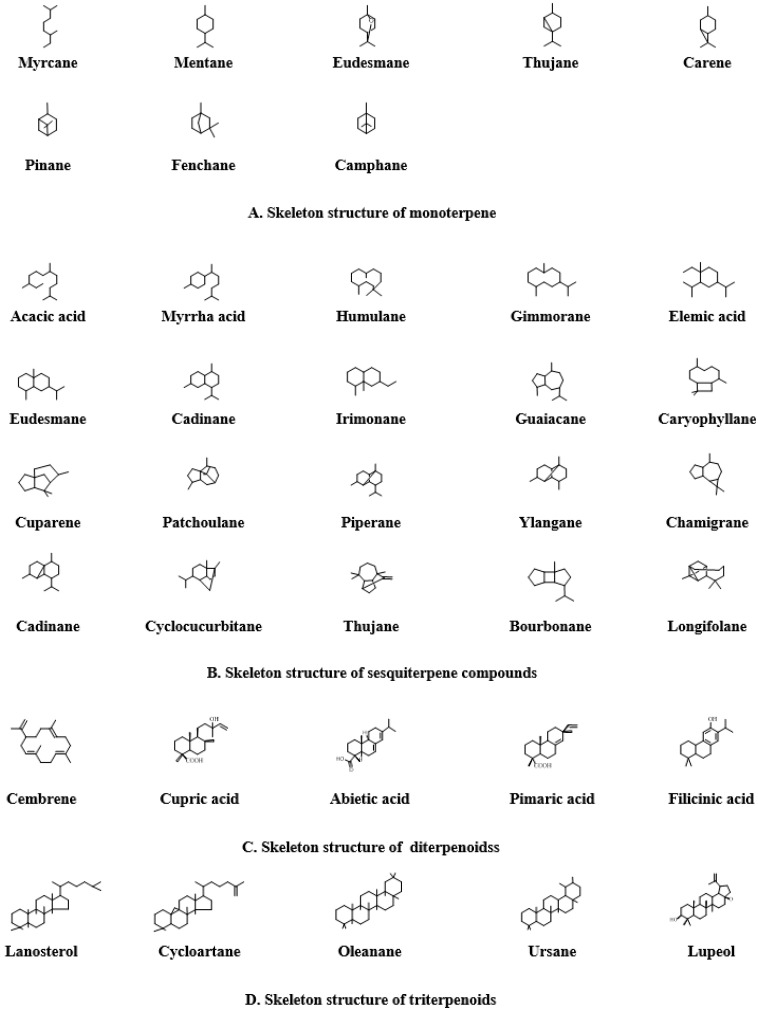
Basic skeleton of terpenoids in propolis. (**A**) Skeleton structure of monoterpene; (**B**) Skeleton structure of sesquiterpene compounds; (**C**) Skeleton structure of diterpenoidss; (**D**) Skeleton structure of triterpenoids.

**Figure 5 nutrients-17-02803-f005:**
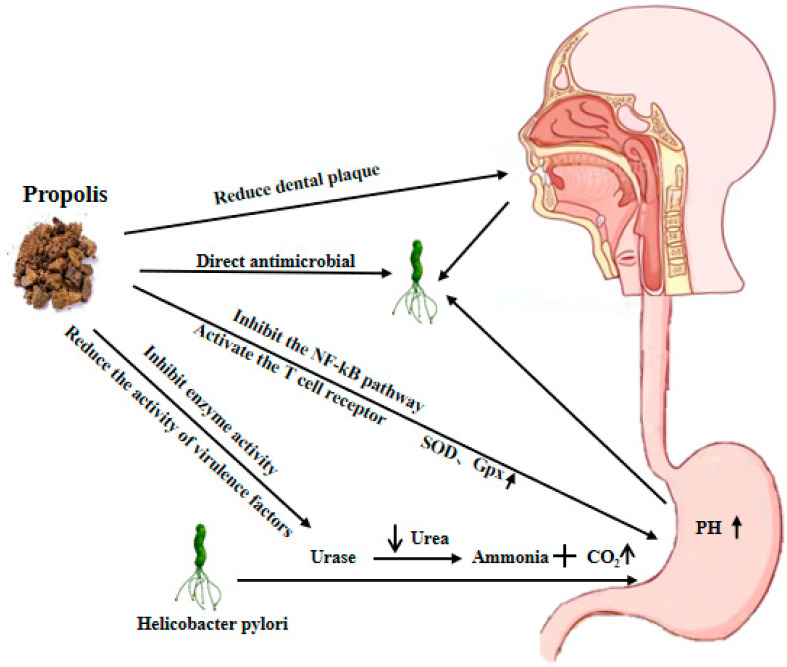
The mechanisms of action of propolis in the process of combating *H. pylori* infection.(Arrows represent the rise or fall of a substance or activity).

## Data Availability

Not applicable.

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
