# Peer review of "Potential Efficacy of Propolis in Treating Helicobacter pylori Infection and Its Mechanisms of Action"

_nutrients, 2025, doi:10.3390/nu17172803_

Round 1
Reviewer 1 Report
Comments and Suggestions for Authors
Dear Editor,
Please, find below a revision of the manuscript entitled: “Potential Efficacy of Propolis in Treating Helicobacter pylori Infection and Its Mechanisms of Action” submitted to the journal - Nutrients.
Helicobacter pylori (H. pylori) is a serious pathogen associated with a variety of gastrointestinal disorders, particularly connected with gastric cancer and rectal tumor. Therefore, a demonstration of propolis activity and its mechanism of action against H. pylori is not only interesting but also scientifically important.
Despite the interesting assumptions, the manuscript preparation is poor. Thus, I propose to accept the manuscript after a major revision, and almost all sections should be considered and rewritten.
- The Introduction section should be better show the main problem of this article.
- In the second section, there is no continuity of information between subsections.
Why did the author show very short subsections containing only short communicating sentences without the development of the mechanisms?
Moreover, the title of this section suggests that the author will present the pathogenic mechanisms of pylori, whereas there is a lack of such data. - In the third section, the author lists chemical compounds within propolis, without consideration of their biological activity.
In this section, the author should show discerning cogitation, according to the new title of the subsection "Propolis, chemical composition, and biological activity". And then, in the fourth section, the mechanism of propolis against pylori should be discussed. - Whereas, in the fourth section, there were only one and a half or two pages on this topic. Moreover, in the fourth section, the author repeats the same information from the Introduction and the second section of the manuscript.
Therefore, in such a form, the manuscript cannot be accepted and should be considerably corrected.
Comments on the Quality of English LanguageThe article needs English revision to improve the consideration of the problem and further discerning discussion.
Author Response
Comments 1:The Introduction section should be better show the main problem of this article.
Response1: The annotated manuscript was very helpful, thank you for your careful review. We have expanded the introduction section to better present the main issues of this article. The following content has been added: Firstly, lines 38-43 emphasize the unique survival of Helicobacter pylori in gastric acid environment and its status as a Class 1 carcinogen; Further elaborate on the resistance challenges and additional side effects faced by current quadruple therapy in provided in lines 44-47 and 53-56; Finally, in lines 56-67 and 69-78, the complex chemical composition, multiple biological activities (especially anti biofilm and anti-inflammatory effects), potential as an alternative or adjuvant treatment strategy, and multi-target mechanisms of propolis were systematically supplemented, providing a more comprehensive theoretical basis for research.
Comments 2:In the second section, there is no continuity of information between subsections.
Why did the author show very short subsections containing only short communicating sentences without the development of the mechanisms?
Moreover, the title of this section suggests that the author will present the pathogenic mechanisms of pylori, whereas there is a lack of such data.
Response 2: We appreciate the constructive suggestions from the reviewers. We have reorganized the logical structure of this section to form a clear 'storyline': for example, in lines 80-99, we have added a paragraph emphasizing the key host signaling pathways (such as CagA, VacA, T4SS, etc.) that Helicobacter pylori colonizes, acts on toxins and effector proteins, and activates, ultimately leading to pathological outcomes (such as inflammation, cell damage, and cancer). We have also added transitional sentences between each section to ensure smooth and logically coherent text.
Secondly, we conducted in-depth exploration of the pathogenic mechanism, rather than just describing the phenotype. For example, lines 129-132 provide a detailed description of how CagA is injected into host cells and interacts with host proteins.
Comments 3:In the third section, the author lists chemical compounds within propolis, without consideration of their biological activity.
In this section, the author should show discerning cogitation, according to the new title of the subsection "Propolis, chemical composition, and biological activity".
Response 3: We appreciate the reviewer's careful review and helpful comments. The suggestion to only list the compounds in propolis without considering their biological activity in the third section is very good. We reorganized the paragraph structure based on this suggestion. In the third part, we first supplement the introduction of what propolis is and its many biological activities (such as antibacterial, antioxidant, etc.) in lines 210-214.
Secondly, we set up a subheading with the structure of active ingredients in propolis as the first subheading, and added a second subheading in lines 262-279. This article mainly provides a detailed introduction to the core biological activities of propolis and which active ingredients work.
Comments 4:And then, in the fourth section, the mechanism of propolis against pylori should be discussed. Whereas, in the fourth section, there were only one and a half or two pages on this topic. Moreover, in the fourth section, the author repeats the same information from the Introduction and the second section of the manuscript.
Response 4: We appreciate the constructive suggestions from the reviewers. Based on your suggestion, we have carefully revised the fourth part and removed the overlapping parts with the introduction and the second part. And the content of the fourth part has been reorganized, correspondingly increasing the length of the mechanism of action of propolis on Helicobacter pylori, consistent with the title, and emphasizing the mechanism of action. For example, lines 313-321 mainly discuss the antibacterial mechanism of propolis, lines 361-366 mainly discuss the antioxidant mechanism of propolis, and so on.
Comments 5:Comments on the Quality of English Language:
The article needs English revision to improve the consideration of the problem and further discerning discussion.
Respond: Thank you for your comment. We have substantially revised the manuscript to correct any grammatical or spelling mistakes. In addition, we had the paper edited by professional English writers to enhance the readability.
Reviewer 2 Report
Comments and Suggestions for Authors
Thank you for the opportunity to review the manuscript entitled “Potential Efficacy of Propolis in Treating Helicobacter pylori Infection and Its Mechanisms of Action”.
This review article explores the potential efficacy of propolis in the treatment of Helicobacter pylori (H. pylori) infections, a bacterium implicated in several gastrointestinal diseases, including gastritis, peptic ulcers, and gastric cancer.
The authors provide a detailed overview of the pathogenic mechanisms of H. pylori, focusing on the limitations of conventional antibiotic therapies, such as increasing antibiotic resistance, side effects, and alterations in the gut microbiota. Propolis, a natural product of bees, presents a promising therapeutic alternative thanks to its antibacterial, anti-inflammatory, antioxidant, and immunomodulatory properties.
Despite minor limitations, such as the lack of clinical standardization and the heterogeneity of propolis composition, the review is scientifically sound, well-referenced.
Comments:
I strongly recommend including a summary table of the main studies cited throughout the manuscript. Such a table would significantly enhance the clarity and accessibility of the review by providing a concise visual overview of key findings. Specifically, it would allow readers to easily compare the types of propolis used (e.g., geographic origin), study models (in vitro, in vivo, clinical), methodologies, and main outcomes. This addition would also strengthen the methodological transparency of the paper.
Author Response
Comments 1:I strongly recommend including a summary table of the main studies cited throughout the manuscript. Such a table would significantly enhance the clarity and accessibility of the review by providing a concise visual overview of key findings. Specifically, it would allow readers to easily compare the types of propolis used (e.g., geographic origin), study models (in vitro, in vivo, clinical), methodologies, and main outcomes. This addition would also strengthen the methodological transparency of the paper.
Respond 1: We sincerely thank the reviewers for their enthusiasm and careful review of our manuscript. We have added a table in lines 439-440 of the main text, which includes propolis types, research models, main methods, key results, and relevant references cited.